# Comparison of Four Electrical Interfacing Circuits in Frequency Up-Conversion Piezoelectric Energy Harvesting

**DOI:** 10.3390/mi13101596

**Published:** 2022-09-26

**Authors:** Han Lu, Kairui Chen, Hao Tang, Weiqun Liu

**Affiliations:** 1School of Intelligent Manufacturing, Panzhihua University, Panzhihua 617000, China; 2School of Mechanical Engineering, Southwest Jiaotong University, Chengdu 610032, China

**Keywords:** piezoelectric vibration energy, frequency up-conversion, interface circuit, coupling level

## Abstract

Efficiently scavenging piezoelectric vibration energy is attracting a lot of interest. One important type is the frequency up-conversion (FUC) energy harvester, in which a low-frequency beam (LFB) impacts a high-frequency beam (HFB). In this paper, four interface circuits, standard energy harvesting (SEH), self-powered synchronous electric charge extraction (SP-SECE), self-powered synchronized switch harvesting on inductor (SP-SSHI) and self-powered optimized SECE (SP-OSECE), are compared while rectifying the generated piezoelectric voltage. The efficiencies of the four circuits are firstly tested at constant displacement and further analyzed. Furthermore, the harvested power under FUC is tested for different electromechanical couplings and different load values. The results show that SP-OSECE performs best in the case of a weak coupling or low-load resistance, for which the maximum power can be 43% higher than that of SEH. As the coupling level increases, SP-SSHI becomes the most efficient circuit with a 31% higher maximum power compared to that of SEH. The reasons for the variations in each circuit with different coupling coefficients are also analyzed.

## 1. Introduction

The continuous development of semiconductors over the last three decades has brought tremendous technological advances in wireless sensor networks (WSNs) [1], microelectromechanical systems (MEMS) [2] and the Internet-of-Things (IoT) [3,4]. Providing an efficient and sustainable power source for these systems without using batteries has become a focus of research [5,6]. There is a multitude of energy in the environment, and vibration energy is one of the most abundant and ubiquitous [7,8]. A piezoelectric harvester that converts the mechanical energy of vibration into electrical energy has been widely studied [9,10,11], due to the high power density of piezoelectric materials.

Generally, a piezoelectric harvester consists of a mechanical oscillator, a piezoelectric transducer unit an electrical energy extraction and a storage unit [12]. In many cases, the resonant frequency of a piezoelectric harvester is relatively high, while the environmental vibration frequency is low. In order to achieve efficient energy harvesting, the resonant frequency of the mechanical oscillator needs to be matched to the low frequency from the environment [13]. Among the methods used so far, increasing the vibration frequency from the environment is an effective solution, namely, the frequency up-conversion (FUC) approach. It has the following advantages: (1) it improves the efficiency of low-frequency vibration energy harvesting, (2) facilitates the expansion of bandwidth, (3) reduces the volume of the harvester and (4) increases the energy density.

The implementation of FUC has been extensively studied. One of the most common structures is the contact up-conversion mechanism, which usually has two or three beams. Gu [14] used three beams to achieve FUC via the periodic impact between the driving beam and the generating beams. Halim et al. [15] presented a low-frequency vibration energy harvester that exploited the mechanical impact of the mass of a flexible dynamic magnifier on a harvester base stopper. This mechanical impact delivers a large secondary force to the secondary beam. Edwards [16] et al. presented a single cantilever FUC mechanism under stochastic excitation when configured as an electromagnetic energy harvester. Wang [17] et al. presented a cantilever beam with magnets attached to the ends that impacted two flexible stoppers, which were placed on each side of the beam. Liu [18] et al. proposed a meandered cantilever as a low-resonant-frequency cantilever and a straight cantilever as a high-resonant-frequency cantilever. Gu [19] et al. proposed an energy harvesting device in which a low-frequency resonator impacted a high-frequency energy harvesting resonator, resulting in energy harvesting predominantly at the system’s coupled vibration frequency. Chen [20] et al. presented the FUC energy harvesting system composed of a lower frequency PZT narrow bimorph with an extended iron proof mass and a higher frequency PZT wide bimorph without a proof mass. Zhang [21] et al. designed a piezoelectric vibration energy harvester in which a high-frequency generating beam was triggered by the rope or impacted directly by the low-frequency driving beam. Halim [22] et al. presented a lateral ball impact that pushed the mass block on top of the parabola and the piezoelectric beam downward, producing vibrations perpendicular to the direction of the ball’s motion. FUC with the contact method also allows ambient excitation to be loaded directly onto the oscillator. Abedini [23] et al. added a pair of rack and pinion and a crank–slider mechanism between the high-frequency beam and the low-frequency beam so that the energy harvester can obtain more energy. In addition to contact up-conversion, magnetically driven non-contact is also available. This type avoids energy loss during contact between the components, reduces noise and increases the energy harvesting range, efficiency and system lifetime. The device consists of a low-frequency piezoelectric cantilever with a rectangular magnet attached to the free end, a higher frequency piezoelectric cantilever with smaller rectangular magnets and an assembled anchor [24,25]. In this mechanism, the driving beam can also be replaced with a disc attached with a magnet, and the rotating disc causes the power generating beam to vibrate [26].

The aforementioned works aimed to optimize the structure of the generator. However, the electricity from the generator is alternating current (AC); in most cases, it cannot directly supply low-power electronic devices. In practical applications, AC is transformed into direct current by an energy extraction circuit. It can also improve the output power. Therefore, a lot of research has been conducted on the optimization of energy extraction circuits. Guyomar [27] et al. proposed a synchronized switch harvesting on an inductor circuit (SSHI). Lefeuvre [28] et al. proposed a synchronous electric charge extraction circuit (SECE). The output power of the SSHI circuit depends on the load. Although the output power of the SECE circuit does not depend on the load of the circuit, it requires a very accurate closing time of the synchronous switch, which is less practical in application. Wu [29] et al., based on the SECE circuit, designed an optimized synchronous charge extraction circuit. Compared with SECE, the circuit changes from one switch to two, the operation mode is more flexible, two diodes are reduced and the power loss is reduced. This circuit has the advantages of the SECE and SSHI circuits, improves the output power of the circuit and has a weak load dependence. The above circuits need to control the opening and closing of one or more switches in the process of extracting energy. The types of switches are divided into electronic switches and mechanical switches [30,31,32,33]. According to the circuit of the control switch, electronic switches can be divided into a self-power supply [34,35,36,37,38] and an external power supply [39,40]. In practical application, it is inconvenient that an external power supply is required for the external power supply. Therefore, the self-power supply has been considerably studied. Self-powered switching circuits, the electronic components powered by the piezoelectric source automatically, detect the piezoelectric voltage maxima or minima and turn the switch on or off properly. Although the electronic switch can stabilize the output power, the electronic components in the circuit will consume some energy. The delayed opening of the electronic switch will reduce the power of the harvester.

At present, most of the research on FUC mainly focuses on the design and improvement of the mechanical structure, while there is less research on the circuit of energy harvesting. The research regarding energy extraction circuits is mainly aimed at the steady-state condition, and FUC is rarely considered. The energy generated under FUC is unstable and attenuated. As a relatively special energy form, it is different from the energy generated at constant displacement and constant force under steady-state conditions. During observation of the energy input mode, the injected energy of each cycle into the FUC mechanism is a constant value, while the injected energy in the case of constant displacement or constant excitation under steady-state conditions varies. At present, there are few studies to compare the energy harvesting performance of the circuit under the condition of FUC. This paper applies SEH, SP-SSHI, SP-SECE and SP-OSECE to harvest the energy generated by the FUC vibration energy harvester and offers a comparative analysis of the differences in the energy scavenged by the four circuits under FUC conditions.

## 2. Theoretical Model

### 2.1. Device Configuration and Working Principle

As shown in Figure 1e, the energy harvester system with frequency up-conversion is composed of two cantilever beams that face each other. The low-frequency beam is narrower and longer, with a mass block at the end. The high-frequency beam is integrated with four parallel-arrayed piezoelectric thin-film energy harvesting elements, and the end is pasted with a mass block. The resonant frequency of the low-frequency beam is 4.85 Hz, and that of the high-frequency beam is 36.25 Hz. There is a distance, *d*, between the low-frequency beam and the high-frequency beam, which affects the impact performance between the two beams.

When the FUC energy harvesting system is excited by the external low-frequency vibration energy, the low-frequency beam resonates and impacts the high-frequency beam via its end mass. Afterwards, the low-frequency beam and the high-frequency beam move together for a short period of time. This is the first stage of mechanical movement. In this stage, the kinetic energy of the low-frequency beam is transferred to the high-frequency beam. In the second stage, the two beams begin to separate and then vibrate independently. The amplitude of the high-frequency beam attenuates exponentially at its resonance frequency, thus converting the vibration energy into electrical energy through the piezoelectric effect. The low-frequency beam continues to be excited by the low-frequency vibration energy in the environment until it hits the high-frequency beam in the next cycle.

### 2.2. Modeling of the FUC Energy Harvesting System

The model of the piezoelectric energy generator based on FUC is shown in Figure 2. The harvesting system can be assumed to be two spring–mass–damping balance models. Here, $k_{0}$, $c_{0}$, $m_{0}$ are the spring stiffness coefficient, damping coefficient and mass of the low-frequency beam in the system. The high-frequency beam with a spring stiffness$\;k_{1}$, damping factor $c_{1}$, and mass $m_{1\;}$is placed at a distance of d from the low-frequency beam. In the model, the displacements of the high-frequency beam and low-frequency beam are $s_{1}\;$and$\;s_{0}$, The second order differential equation of the mass motion in phase 2 can be expressed as follows [18,21]:(1)$$m_{0}\ddot{s_{0}}+c_{0}\dot{s_{0}}+k_{0}s_{0}=-m_{0}a\sin\left(\omega t\right)\;;\;s_{0}-s_{1}<d$$
(2)$$m_{1}\ddot{s_{1}}+c_{1}\dot{s_{1}}+k_{1}s_{1}=-m_{1}\mathrm{asin}\left(\omega t\right)\;;\;s_{0}-s_{1}<d$$
where *a* and ω are, respectively, the acceleration of excitation and the circular frequency of excitation, and the time is expressed as *t*.

In phase 1, as shown in case 2, the mass motion equation of the high-frequency beam and the low-frequency beam combined is shown as follows by assuming that the impact loss is included in the damping ratio [41,42]:(3)$$\left(m_{0}+m_{1}\right)\ddot{s_{0}}+(c_{0}+c_{1}\dot{)s_{0}}+k_{0}s_{0}+k_{1}\left(s_{0}-d\right)=-\left(m_{0}+m_{1}\right)\mathrm{asin}\left(\omega t\right)\;;\;s_{0\;}-s_{1}\geq d\;$$

### 2.3. Voltage Model

The open-circuit voltage generated by the high-frequency beam can be calculated according to the displacement $X_{0}$ [19,43]. The calculation formula is as follows:(4)$$V_{o}=-\frac{d_{31}t_{p}\sigma_{s}}{\varepsilon }=\left(-\frac{d_{31}t_{p}}{\varepsilon }\right)\frac{3X_{0}L_{p3}}{b_{p}t_{p}{}^{2}}$$
where $d_{31}$ is the piezoelectric coefficient, $t_{p}$ is the thickness of the high-frequency beam, ε is dielectric constant, $\sigma_{s}$ is the stress distribution on a high frequency and $L_{p3}\;\mathrm{and}\;b_{p}$ are the length and width of the high frequency, respectively.

According to the two dynamic stages of energy acquisition described above, the generated voltage can be written as a function of time:(5)$$V=\{\begin{array}{c} \frac{V_{0}}{2}\sin\left(\omega_{coup}t\right)e^{-\xi_{t1}\omega_{coup}t}\;,\;n\frac{2\pi }{\omega_{0}}<t<\left(n+1\right)\frac{3\pi }{2\omega_{coup}};\;phase1\\ \frac{V_{0}}{2}\sin\left(\omega_{1}\left(t-\left(n+1\right)\frac{3\pi }{2\omega_{coup}}\right)+\frac{3\pi }{2}\right)e^{-\xi_{t2}\omega_{1}t}\;,\;\\ \left(n+1\right)\frac{3\pi }{2\omega_{coup}}\leq t\leq \frac{\pi }{2\omega_{0}}\left(n+1\right);\;phase2 \end{array}$$
$$\xi_{t1}=\frac{c_{0}}{2(m_{0}+m_{1})\omega_{0}}+\frac{c_{1}}{2(m_{0}+m_{1})\omega_{1}};\xi_{t2}=\frac{c_{1}}{2m_{1}\omega_{1}};\omega_{0}{}^{2}=\frac{k_{0}}{m_{0}};\omega_{1}{}^{2}=\frac{k_{1}}{m_{1}};\omega_{coup}=\sqrt{\frac{k_{0}+k_{1}}{\left(m_{0}+m_{1}\right)}}$$
where *n* is the number of cycles of the collector,$\;\xi_{t1}\;$is the total damping ratio of the coupled vibration, $\xi_{t2}$ is the total damping ratio of the free vibration of the high-frequency beam, $\omega_{1}$ is the resonant frequency of the high-frequency beam, $\omega_{0}$ is the resonant frequency of the low-frequency beam and $\omega_{coup}$ is the resonance frequency in coupled motion.

### 2.4. Energy Extraction Circuit

In the piezoelectric energy harvesting system, the piezoelectric energy extraction circuit is an indispensable part that also affects the energy harvesting efficiency. The principle of the standard energy harvesting interface (SEH) [41,43] is shown in Figure 1a, which mainly consists of a full-bridge rectifier composed of four diodes and a smoothing capacitor (C_r_). The load resistance (R_load_) represents the equivalent input resistance of the following electronic module to be supplied. The diode is equivalent to an ideal diode, and the voltage drop V_D_ and its loss are ignored. When V_P_ > V_load_, the SEH circuit can transfer the energy generated by the piezoelectric element to the load end. The synchronous electrical charge extraction [28,44,45], as shown in Figure 1b, is an improvement of the standard circuit, which adds an inductance L, a switch S and a diode D between the rectifier bridge and the filter capacitor. In addition, a peak detection circuit is added to control the switch. The peak detection circuit is composed of an envelope detection circuit (1) and a comparator circuit (2). When the switch S is closed, L-C_0_ circuit oscillation is established, and the charge accumulated in the clamped capacitor C_0_ is transferred to the inductor L. After one quarter of the oscillation cycle, the energy stored on the inductor is transferred to the load once the switch is opened. The left side of the transformer is the primary-side coil, and the right side is the secondary-side coil. Parallel synchronized switch harvesting on inductor [46,47] is shown in Figure 1c. Compared with the standard circuit, an inductance, two electronic switches S and two identical peak detection circuits are added. The peak detection forces the corresponding switch to open when the output voltage of the piezoelectric element is the maximum value of the positive half cycle and to close when it is the minimum value of the negative half cycle. The peak circuit on the right is opposite to that on the left. When the mechanical displacement reaches the maximum and minimum values, the two switches S are closed alternately to establish L-C_0_ circuit oscillation. After the switch is closed for half an oscillation period, the piezoelectric voltage is inverted. The optimized synchronous electrical charge extraction is shown in Figure 1d. The flyback transformer is used to replace the inductor. The transformer divides the circuit into two parts. The left part is similar to the parallel-switch circuit, in which the peak detection circuit and the electronic switch circuit are the same as the parallel-switch circuit. The left part of the transformer is the primary-side coil, and the right part is the secondary-side coil. The two switches are closed alternately in a vibration cycle. This switching strategy allows the voltage to be reversed twice in a vibration cycle. The power calculations of the four circuits are shown in Table 1.

## 3. Experimental Results

### 3.1. Experimental Setup

The mechanical part of the experimental system under steady-state conditions and FUC are shown in Figure 3a,b, respectively. The mechanical part includes an aluminum alloy low-frequency beam (110 mm × 10 mm × 0.3 mm) with a low natural frequency and a beryllium copper high-frequency beam (100 mm × 30 mm × 1 mm) with a high natural frequency. The root of the beryllium copper is attached with piezoelectric elements, and the end is pasted with a mass block (24 g). Similarly, the end of the low-frequency beam is pasted with a mass block (54 g). The mechanical part of the energy harvesting device is installed on a vibrating table, which is driven by a signal generator (dg1032, RIGOL, RIGOL Technologies, Inc., Beijing, China) and a power amplifier. The frequency of its sinusoidal excitation can be adjusted. With the adopted laser displacement sensor (HLC203BE, SUNX, Panasonic Industrial Devices SUNX Suzhou Co., Ltd., Suzhou, China), the displacement amplitude of the beam tip is kept constant by tuning the excitation level. An acceleration sensor (PCB©, M352C68, PCB Piezotronics, Inc., New York, USA) is attached to the shaker to acquire the excitations. Meanwhile, the piezoelectric voltage is captured by the oscilloscope while the load voltage is recorded by a voltage meter. Figure 3c shows the printed circuit board used in the experiment, and the circuits on it are the SP-SECE circuit, SP-OSECE circuit, SP-SSHI circuit and SEH circuit, respectively. The used components and their characteristics are listed in Table 2 from their experimental identifications or factory datasheets. The system parameters in the experiment are listed in Table 3.

### 3.2. Experiments under Steady-State Conditions

The efficiency of the circuit affects the energy harvesting of the circuit, so it is necessary to understand the characteristics of the four circuits before they are applied to the FUC generator. As depicted in Figure 4, with the increasing load resistance, the efficiency of the SP-OSECE, SP-SECE and SP-SSHI circuits $\eta =E_{load}/E_{in}$ decreases continuously.${\;E}_{in}$ is the extracted energy from the piezoelectric material, and $E_{load}$ is the energy obtained on the load. Because the energy extracted from the piezoelectric material by these three circuits dissipates on components such as diode D, switch S, transformer L and resistor R, their efficiencies are lower than that of the SEH circuit, and, with the increase in resistance, the more energy that is extracted from the piezoelectric material, the more energy that is dissipated. For a small resistance value, the energy loss of SP-OSECE is mainly located at the secondary-side of the transformer. With the increase in resistance value, more energy is extracted, but the dissipation caused by the circuit quality factor *Q_I_* and the diode connected to the primary-side coil become larger. In addition, the charge neutralization effect is increasingly obvious, and the capacitance value of the piezoelectric elements used is small, resulting in lower and lower efficiencies of the SP-OSECE circuit. The dissipation of SP-SECE is mainly caused by the full-bridge circuit and the circuit quality *Q_I_*, which have a relatively small growth range along with an increasing resistance. From the perspective of circuit structure, SP-SSHI has no secondary-side coils and diodes. Therefore, the energy dissipation of the SP-SSHI circuit is less than that of SP-SECE and SP-OSECE with a higher efficiency. In the steady state, in order to avoid the influence of electromechanical coupling and to focus on the circuits, the constant displacement case is considered first with the tip displacement of the high-frequency beam kept at 1 mm, while the resistance changes from 10 KΩ to 2 MΩ. The output power results are shown in Figure 5a. The SP-SSHI circuit is the best when it is above 100 KΩ, and the SP-OSECE circuit is the best when it is below 100 KΩ. Their general trends are basically the same, but there is a slight difference. From Figure 5a, it can be seen that the power of SP-OSECE with a large resistance at constant displacement decreases faster than that of the constant force. This is because more energy is extracted from the piezoelectric elements with a large resistance at constant displacement, but the dissipation is also large. As shown in Figure 5b, in the case of constant force, the coupling level decreases, and SP-OSECE becomes outstanding.

### 3.3. Experiments under Frequency Up-Conversion Conditions

The experimental results of FUC are shown in Figure 6. In this figure, points A to B are the coupled motion of the two beams, and points C to D are the free vibration stage of the high-frequency beams.

In the FUC experiment, $k^{2}Q_{m}=0.63\;$and an acceleration of 0.4 g were used to compare the SEH, SP-SECE, SP-SHHI and SP-OSECE circuits. The output waveforms of the piezoelectric elements are shown in Figure 7, and the comparison of the output power is shown in Figure 8. The resistance changes from 10 KΩ to 2 MΩ. If the load of the circuit is below 100 K, the SP-OSECE circuit is recommend. The SP-SSHI circuit can be selected above 100 K. From the results, it can be seen that under the FUC conditions, the SP-SSHI and SEH circuits also have the problem of matching the best resistance, and higher energy can be harvested near the best resistance. SP-SECE has small power variation due to its small dependence on resistance. Since the efficiency of SP-OSECE decreases with an increase in the resistance value, the energy harvesting is highest for small resistances compared to other circuits.

In order to observe the influence of electromechanical coupling on energy harvesting, experiments were carried out on the FUC harvester for several cases of $k^{2}Q_{m}$ = 0.1088, 0.1472, 0.192 and 0.3072. The output power comparison is shown in Figure 9, and the maximum power under different coupling levels is shown in Figure 10. In general, the power obtained by the four circuits increases with the increase in the coupling level because the mechanical energy also increases when it is converted into electrical energy. When the coupling coefficient $k^{2}Q_{m}\leq 0.192$, the SP-OSECE circuit obtains the highest power. For $k^{2}Q_{m}>0.3$, SP-SSHI appears to be the most effective technique for power, when it is around the optimal load. When the electromechanical system is weakly coupled, less mechanical energy is converted into electric energy. Hence, SP-SSHI and SEH extract less energy at this time, and SEH performs worse. However, SP-OSECE has the effect of gaining extracted energy, so it is the best electric circuit in the case of weak coupling. With stronger electromechanical coupling, the energy loss in the SP-OSECE circuit becomes larger, and the efficiency decreases continuously. Therefore, for $k^{2}Q_{m}\;>\;0.63$, SP-OSECE becomes the worst. With the increase in the coupling level, the energy extracted by the SP-SSHI circuit increases, and its efficiency decreases less than that of SP-OSECE and SP-SECE, so the load end of the SP-SSHI circuit obtains the highest power. The reduction in the efficiency of the SP-SECE circuit is less than that of SP-OSECE, which results in the power obtained at the load end being slightly higher than that of SP-OSECE after the coupling level increases. Although the SEH circuit itself extracts less energy from the piezoelectric element, it has a higher efficiency due to less dissipation. It improves in the case of strong coupling. If the coupling level continues to increase, the conversions of mechanical energy into electrical energy by the other three circuits will be suppressed. When the coupling coefficient reaches a certain value, SEH will be the best circuit. Experiments with higher coupling coefficients were not implemented due to experimental conditions.

## 4. Conclusions

In this paper, a typical FUC device is described, which is composed of two beams. There are mass blocks at the end of the low-frequency beam and the end of the high-frequency beam. Four circuits, SP-OSECE, SP-SECE, SP-SSHI and SEH, were used to harvest energy with the FUC generator for comparison purposes. The efficiency of the four circuits was firstly tested in steady state with a constant displacement of 1 mm. The results show that the efficiency of SP-OSECE decreases with an increasing resistance value, while SP-SECE has a smaller reduction, and SP-SSHI and SEH are more efficient. Especially at large resistances, SP-SSHI works more efficiently than SP-OSECE, with more than two times the efficiency, while SEH performs more than three times as efficiently as SP-OSECE. The reasons for the decrease in efficiency as the resistance value increases and the difference in efficiency between the four types of circuits were analyzed. The experiments on the FUC vibration energy harvesting were conducted with different $k^{2}Q_{m}$. Based on the findings of the experiments, selecting the best circuit requires comprehensive consideration of the magnitude of load resistance and the strength of the electromechanical coupling. It is recommended that the SP-OSECE circuit is suitable when the electromechanical coupling is weak or the load value is low. Under these situations, the maximum power of the SP-OSECE circuit can be 43%, 15% and 40% higher than those of SEH, SP-SSHI and SP-SECE, respectively. In the case of moderate electromechanical coupling, the SP-SSHI circuit can harvest the most power around its optimal resistance. Specifically, the maximum power of SP-SSHI can be 31% higher than that of SEH, and the improved values will become 24% and 21% when compared with SP-SECE and SP-OSECE.

## Figures and Tables

**Figure 1 micromachines-13-01596-f001:**
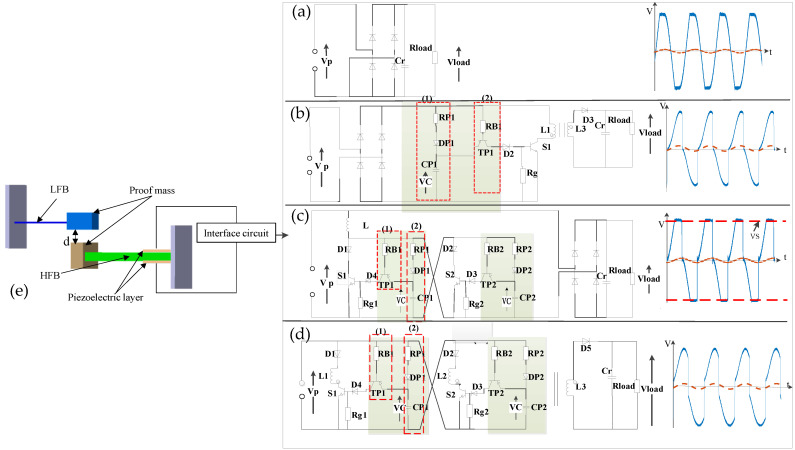
Configuration of the FUC energy harvesting system. (**a**) SEH circuit and wave of piezoelectric voltage (**b**) SP-SECE circuit and waveforms of piezoelectric voltage (**c**) SP-SSHI circuit and waveform s of piezoelectric voltage (**d**) SP-OSECE circuit and waveforms of piezoelectric voltage (**e**) Mechanical structure of FUC.

**Figure 2 micromachines-13-01596-f002:**
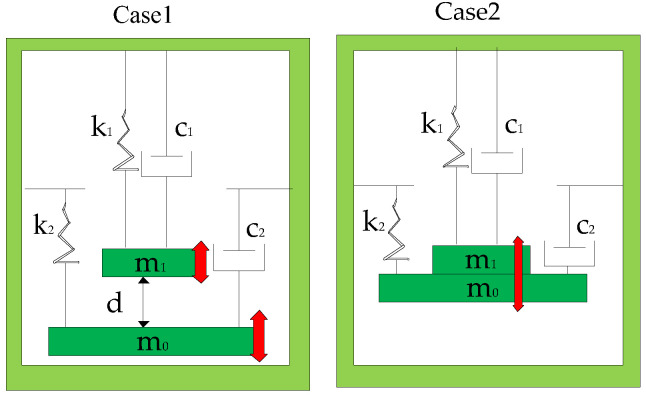
Modeling of cases 1 and 2 of FUC system.

**Figure 3 micromachines-13-01596-f003:**
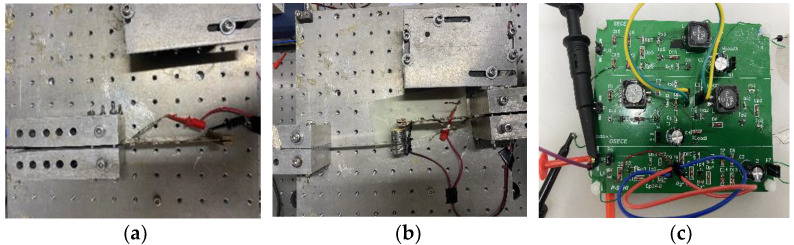
(**a**) The mechanical part of the experimental system for steady-state conditions; (**b**) the mechanical part of the experimental system for frequency up-conversion; (**c**) interface circuit of experimental systems.

**Figure 4 micromachines-13-01596-f004:**
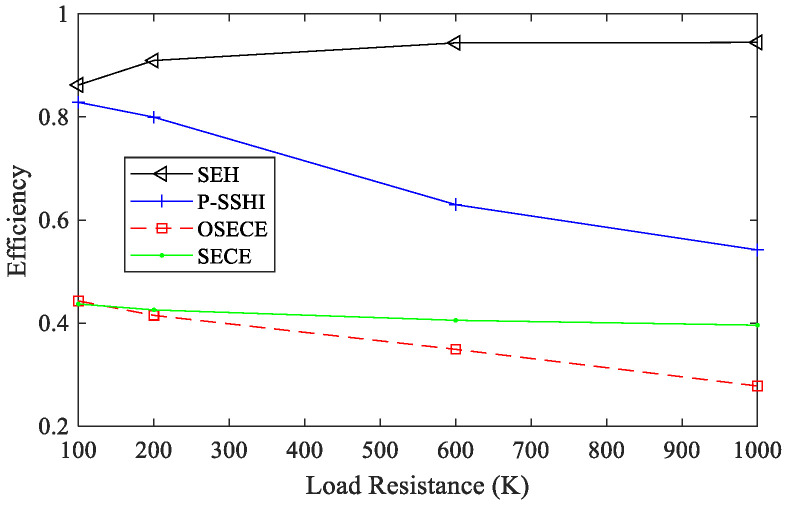
Efficiency of different loads under two piezoelectric plates at constant displacement of 1 mm ($k^{2}Q_{m}=0.63$).

**Figure 5 micromachines-13-01596-f005:**
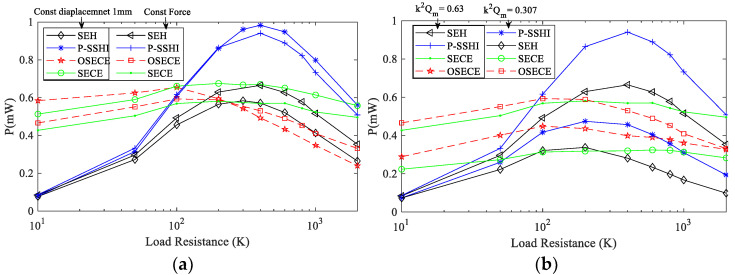
(**a**) Power of different loads at constant displacement of 1 mm and constant force ($k^{2}Q_{m}=0.63$); (**b**) power of different loads at $k^{2}Q_{m}=0.63$ and$\;k^{2}Q_{m}=0.3072$.

**Figure 6 micromachines-13-01596-f006:**
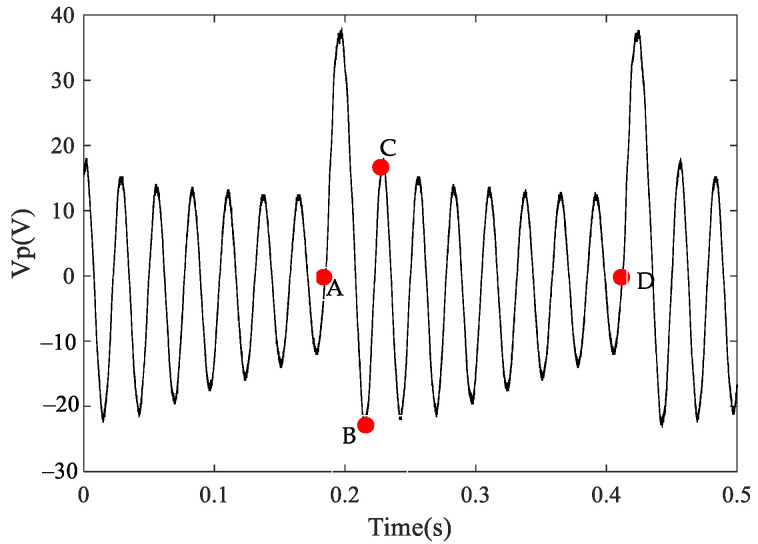
The open–circuit voltages of the HFB.

**Figure 7 micromachines-13-01596-f007:**
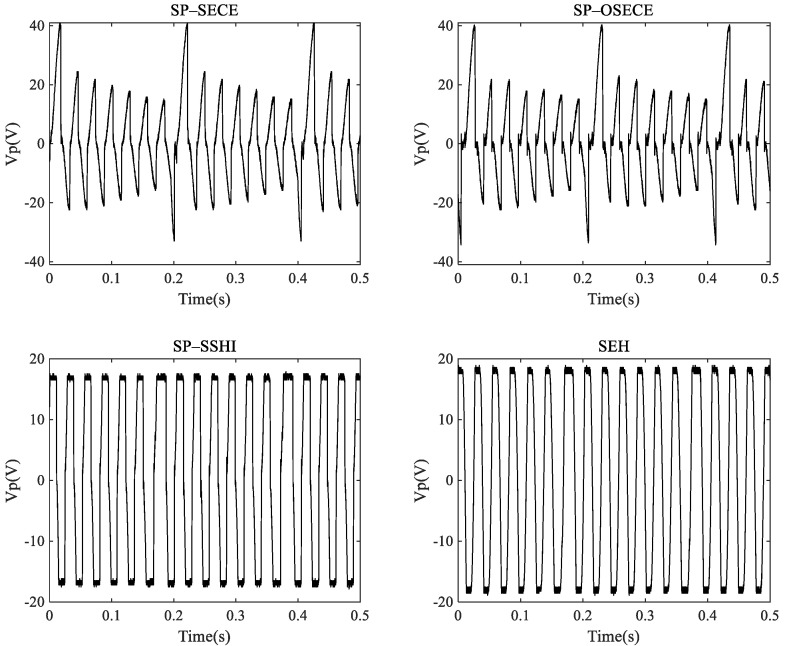
Experimental waveforms of the piezoelectric harvester ($k^{2}Q_{m}=0.63$).

**Figure 8 micromachines-13-01596-f008:**
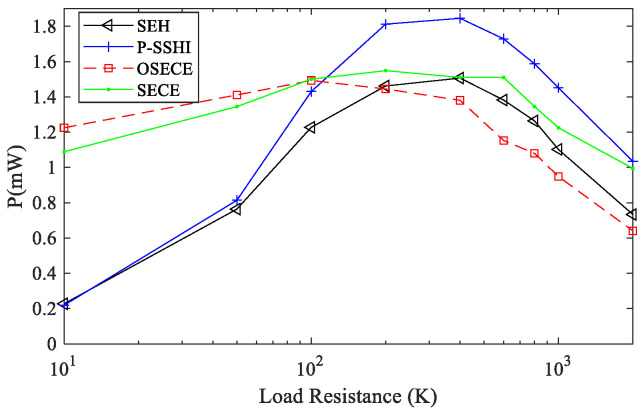
Power of different loads during frequency up-conversion($k^{2}Q_{m}=0.63$).

**Figure 9 micromachines-13-01596-f009:**
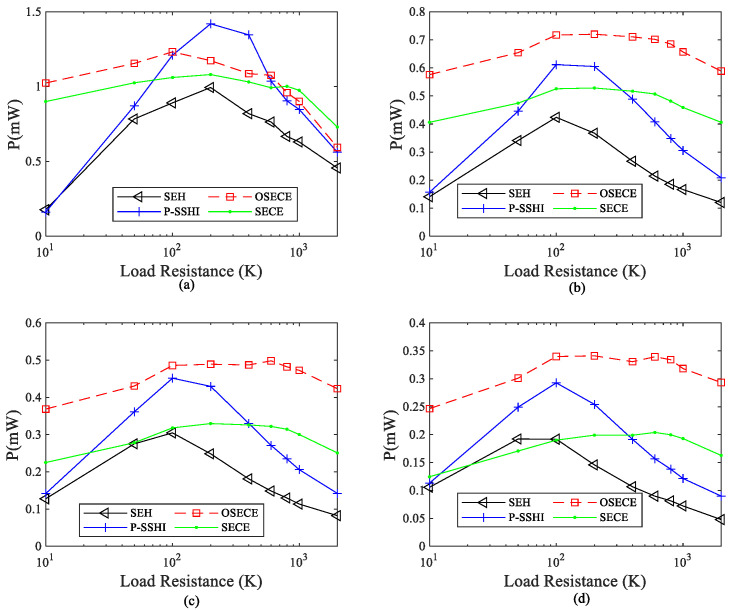
Power under different electromechanical coupling coefficients. (**a**) Coupling coefficient $k^{2}Q_{m}$ = 0.3072; (**b**) coupling coefficient $k^{2}Q_{m}$ = 0.192; (**c**) coupling coefficient $k^{2}Q_{m}$ = 0.1472; (**d**) coupling coefficient $k^{2}Q_{m}$ = 0.1088.

**Figure 10 micromachines-13-01596-f010:**
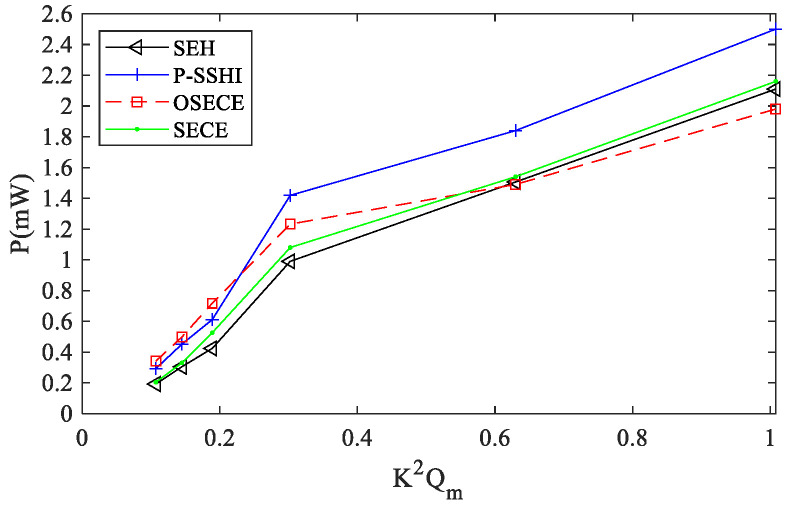
Harvested power against different $k^{2}Q_{m}$.

**Table 1 micromachines-13-01596-t001:** Normalized harvested power for different interfaces [48,49].

Interface	Harvested Power
SEH	$P=\frac{\pi k^{2}Q_{m}}{{\left[\frac{\pi }{4}+\frac{\zeta_{R}}{{\left(1+\zeta_{R}\right)}^{2}}k^{2}Q_{m}\right]}^{2}}\frac{\zeta_{R}{}^{*}}{{\left(1+\zeta_{R}\right)}^{2}}$
SP-SSHI	$P=4f_{0}C_{0}V_{S}\left[V_{P}-V_{D}+V_{D}e^{-\pi /2Q_{I}}\right]-2f_{0}C_{0}V_{S}{}^{2}\left(1-e^{-\pi /2Q_{I}}\right)$
SP-SECE	$P=\frac{\pi k^{2}Q_{m}}{{\left[\frac{\pi }{4}+k^{2}Q_{m}\right]}^{2}}e^{-\pi /2Q_{I}}$
SP-OSECE	$P=\frac{\pi k^{2}Q_{m}}{{\left[\frac{\pi }{4}+Xk^{2}Q_{m}\right]}^{2}}\frac{sin^{2}\left(\omega_{I}t_{m}\right)e^{\frac{-\omega_{I}t_{m}}{Q_{I}}}}{{\left[1+\varepsilon_{c}+\cos\left(\omega_{I}t_{m}\right)e^{\frac{-\omega_{I}t_{m}}{2Q_{I}}}\right]}^{2}}\frac{{\left(2-2\varepsilon_{v2}-\varepsilon_{v2}\varepsilon_{c}\right)}^{2}}{4}$

* $\zeta_{R}=2R_{l}C_{p}\omega /\pi$, $Q_{I}$ is the quality factor of the L-C_0_ oscillating circuit, $k^{2}Q_{m}$ is the figure of merit of the electromechanical structure, $\varepsilon_{v2}$ is voltage ratio and $\omega_{I}t_{m}=\arctan\left(-\frac{2}{\sqrt{\zeta_{R}}}\right)$+π, $\varepsilon_{c}=\frac{C_{p}}{C_{0}}$.

**Table 2 micromachines-13-01596-t002:** Components and parameters.

Definition	Value	Definition	Value
Diode (D_i_, D_pi_)	BAQ135	C_r_	100 uF
Transformer	MSD1278T-105KL	R_bi_	3.3 kΩ
BJT (S_i_)	MMBTA05LT1G	R_gi_	1 MΩ
R_load_	Pin-type resistance (external connection)	R_pi_	100 kΩ
Transistor (T_pi_)	MMBTA56	C_pi_	1 nf
Inductance L_1_, L_2_, L_3_ (H)	1 × 10^−3^		

**Table 3 micromachines-13-01596-t003:** System parameters in the experiment.

Parameter	Value
Clamped capacitance of the piezoelectric element C_0_ (F)	20.68 × 10^−9^
Piezoelectric coefficient α (N/V)	5.557 × 10^−4^
Open-circuit mechanical quality factor Q_m_	63.8
Squared electromechanical coupling coefficient k^2^	0.01
Open-circuit resonance frequency of HFB f_1_ (Hz)	36.25
Stiffness of high-frequency beams (N/m)	1450

## Data Availability

Not applicable.

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
