# Peer review of "Comparison of Four Electrical Interfacing Circuits in Frequency Up-Conversion Piezoelectric Energy Harvesting"

_micromachines, 2022, doi:10.3390/mi13101596_

Round 1

Reviewer 1 Report

This paper studied the comparison of four interface circuits for piezoelectric energy harvesting with frequency up-conversion means. A typical FUC device is elaborated which is composed of two beams. Four circuits, SP-OSECE, SP-SECE, SP-SSHI and SEH, are used to harvest energy. The efficiency of the four circuits was firstly tested in steady state with a constant displacement. The SP-SSHI and SEH are more efficient. The reasons for the decrease in efficiency as the resistance value increases and the difference in efficiency between the four types of circuits are analyzed. Furthermore, the harvested power under FUC is tested for different electromechanical coupling and different load values. The results show that SP-OSECE is best for the cases of weak coupling or low load resistance value. As the coupling level increases, SP-SSHI is recommended for the best power performance. This is a carefully done study and the finding are of considerable interest. However, some concerns should be addressed as follow:
1. Some symbols in Table1 are not specified for physical significance. They are supposed to be indicated for better readability and understanding.
2. The Figure7 contains four subfigures, so the subfigure should also contain the corresponding subtitles.
3. Some expression involved experiment results should use past tense, such as content in line 337-338.
4. The letter subscripts in Tables 2 and 3 do not conform to the specifications.
5. It would be better to use FUC instead of frequency up-conversion in line 105 and 135, harvester instead of collector in line 37, harvested instead of captured in line 286.

Reviewer 2 Report

There is a recorded memo throughout the manuscript. Thus, the authors should modify this one. 

Except for that part, the paper is well organized and enough to be published in Micromachines.

Please note that the recent research trend of piezoelectric energy harvesting (PEH) is a phononic crystal or metamaterial-based PEH. Therefore, several papers are encouraged to be introduced in the Introduction.

[R1]  Lee, Geon, et al. "Piezoelectric energy harvesting using mechanical metamaterials and phononic crystals." Communications Physics 5.1 (2022): 1-16.

[R2] Shao, Hanbo, Guoping Chen, and Huan He. "Elastic wave localization and energy harvesting defined by piezoelectric patches on phononic crystal waveguide." Physics Letters A 403 (2021): 127366.

[R3] Jo, Soo-Ho, and Byeng D. Youn. "A phononic crystal with differently configured double defects for broadband elastic wave energy localization and harvesting." Crystals 11.6 (2021): 643.

[R4] Wen, Zhihui, et al. "Topological cavities in phononic plates for robust energy harvesting." Mechanical Systems and Signal Processing 162 (2022): 108047.
